# Using Copper-Doped Mesoporous Bioactive Glass Nanospheres to Impart Anti-Bacterial Properties to Dental Composites

**DOI:** 10.3390/pharmaceutics14102241

**Published:** 2022-10-20

**Authors:** Arooj Munir, Danijela Marovic, Liebert Parreiras Nogueira, Roger Simm, Ali-Oddin Naemi, Sander Marius Landrø, Magnus Helgerud, Kai Zheng, Matej Par, Tobias T. Tauböck, Thomas Attin, Zrinka Tarle, Aldo R. Boccaccini, Håvard J. Haugen

**Affiliations:** 1Department of Biomaterials and Oral Research Laboratory, Institute of Clinical Dentistry, Faculty of Dentistry, University of Oslo, 0317 Oslo, Norway; 2Department of Endodontics and Restorative Dentistry, School of Dental Medicine, University of Zagreb, 10000 Zagreb, Croatia; 3Institute of Oral Biology, Faculty of Dentistry, University of Oslo, 0317 Oslo, Norway; 4Jiangsu Province Engineering Research Center of Stomatological Translation Medicine, Nanjing Medical University, Nanjing 210029, China; 5Department of Conservative and Preventive Dentistry, Center for Dental Medicine, University of Zurich, 8032 Zurich, Switzerland; 6Department of Materials Science and Engineering, Institute of Biomaterials, University of Erlangen-Nuremberg, 91054 Erlangen, Germany

**Keywords:** dental, resin composites, copper, bioactive glass, mesoporous, nanoparticle, anti-bacterial

## Abstract

Experimental dental resin composites containing copper-doped mesoporous bioactive glass nanospheres (Cu-MBGN) were developed to impart anti-bacterial properties. Increasing amounts of Cu-MBGN (0, 1, 5 and 10 wt%) were added to the BisGMA/TEGDMA resin matrix containing micro- and nano-fillers of inert glass, keeping the resin/filler ratio constant. Surface micromorphology and elemental analysis were performed to evaluate the homogeneous distribution of filler particles. The study investigated the effects of Cu-MBGN on the degree of conversion, polymerization shrinkage, porosity, ion release and anti-bacterial activity on *S. mutans* and *A. naeslundii*. Experimental materials containing Cu-MBGN showed a dose-dependent Cu release with an initial burst and a further increase after 28 days. The composite containing 10% Cu-MBGN had the best anti-bacterial effect on *S. mutans*, as evidenced by the lowest adherence of free-floating bacteria and biofilm formation. In contrast, the 45S5-containing materials had the highest *S. mutans* adherence. Ca release was highest in the bioactive control containing 15% 45S5, which correlated with the highest number of open porosities on the surface. Polymerization shrinkage was similar for all tested materials, ranging from 3.8 to 4.2%, while the degree of conversion was lower for Cu-MBGN materials. Cu-MBGN composites showed better anti-bacterial properties than composites with 45S5 BG.

## 1. Introduction

Dental caries is one of the most common dental diseases affecting more than one-third of the population worldwide [1]. Recently, several restorative biomaterials, including resin-composite-based materials, have been designed to repair damaged tissues [2]. Dental resin composites restore tooth substance lost due to caries and have satisfactory mechanical properties [3]. However, they do not have acid-neutralizing or anti-bacterial properties. Thus, they do not protect surrounding dental tissues from further decay. As a result, secondary caries may occur at the cavity margins, especially in individuals with a high caries risk [4]. Secondary caries is usually attributed to microgaps at the marginal imperfection of restoration, which causes bacterial invasion and demineralization. Secondary caries is a major cause of dental restoration failure, which imposes a significant economic burden on individuals and governments [5].

Researchers are incorporating compounds such as amorphous calcium phosphate (ACP) and bioactive glass (BG) to counter these challenges in dental resins. BG has attracted much attention in the field due to its flexible composition, which can be adjusted to attain a desired biological and remineralization response. Some studies have shown that incorporating ACP [6,7,8,9] and the well-known 45S5 BG [10,11,12,13,14,15,16,17] into composite materials can impart remineralizing properties to the demineralized tooth structure. ACP releases high calcium and phosphate ion concentrations that induce hydroxyapatite formation and enamel remineralization [18]. However, ACP fails to provide high mechanical properties [19,20,21], and highly concentrated BG-based resins reduce mechanical properties and the degree of conversion [22,23].

Commercially available composites use silane coupling agents to improve mechanical strength and prevent crack propagation after the material has been subjected to high stress. Silane coupling molecules are bifunctional. They form a covalent bond (Si-O-Si) with the filler particles on one side and simultaneously copolymerize with the methacrylate group in the resin phase. This helps to prevent the propagation of cracks and increases the mechanical strength of resin composites [24]. However, the silane layer enveloping the bioactive filler prevents ion release, affecting the fillers’ required reactivity [19,21,24]. To overcome this limitation, micromechanical bonding could replace the chemical bonding of silane. Therefore, mesoporous bioactive glasses (MBGs) could be an ideal alternative. Due to their porous nature, mesoporous particles have a larger surface area and can provide enhanced bioactivity [25]. In addition, a low-viscosity resin can penetrate these mesoporous pores and therefore improve the mechanical bonding of the filler particles into the matrix after polymerization [26]. MBGs are the focus of biomedical research by many groups due to their favorable physiochemical properties [27], the release of calcium, phosphorus, and silicon ions [28], and the ability to introduce additional functionalities by adding metal ions such as copper (Cu) [29,30,31,32] or silver (Ag) [33].

Copper is well known for its multifunctional biological activities [34]. Depending on the dose, copper may exert bactericidal, angiogenic, or pro-osteogenic activities. Therefore, the release of copper from the bioactive glass must be carefully controlled to achieve the particular therapeutic effect of copper [35]. To solve this problem, Zheng et al. synthesized copper containing mesoporous bioactive glass nanospherical (Cu-MBGN) particles of 100–300 nm with a pore size of 2–10 nm, by a sol–gel method. In addition, they used a Cu/L-ascorbic acid complex as a precursor of Cu and reported a tunable concentration of copper ions [31]. Several studies described that synthesized Cu-MBGN has excellent potential to improve anti-bacterial properties [25,36,37] and hydroxyapatite crystal formation [31]. In our previous work, Marovic et al. [38] reported that incorporating Cu-MBGN and silanized microfillers increases the stiffness and hardness of the composite while decreasing its strength. Trimodal CuBG composites with silica and microfillers have durable strength after 28 days, improved elastic modulus and hardness, and high polymerization efficiency [38]. Finally, the follow-up study confirmed the advantages of the trimodal system with reduced polymerization stress. In addition, we found that MBGNs disperse photons in multiple directions, resulting in diffuse backscattering and increased light absorption, resulting in rapid polymerization [39].

While previous studies focused on mechanical [38] and polymerization [39] properties, this study focuses on biological effects. Considering the beneficial effects of the combination of Cu-MBGN and silica nanofillers, we continued the same experimental setup [38] and divided the composites into two groups depending on the number of fillers:Binary filler blend consisting of Ba-glass microfillers as a base and 10 wt% of either Cu-MBGN, silica nanofillers, or 45S5 BG with a total filler amount of 65 wt%.Ternary filler blend consisting of Ba-glass microfillers as a base and a combination of 1, 5 or 10 wt% Cu-MBGN with silica filler up to 15 wt%, making a total filler amount of 70 wt%. As a bioactive control, 15 wt% 45S5 was added to the Ba-glass microfillers (15-BG) and as an inert control, 15 wt% silica fillers were added to the filler base of Ba-glass microfillers.

Therefore, we hypothesized that incorporating Cu-MBGN or the combination of Cu-MBGN and silica nanofillers into the resin matrix could reduce polymerization shrinkage, achieve anti-bacterial properties, reduce biofilm formation, and promote remineralization. In order to verify the hypothesis, we evaluated the micromorphology and uniform dispersion of filler particles in the resulting experimental composites by scanning electron microscopy with energy dispersive X-ray analysis (SEM/EDX). The effect of resin composites with Cu-MBGN on polymerization (degree of conversion, volumetric shrinkage, porosity) was investigated using attenuated total reflectance accessory on a Fourier transform infrared spectroscopy (ATR-FTIR) and micro computed tomography (micro-CT). Remineralizing and anti-bacterial properties were estimated from the release of calcium and copper ions by inductively coupled plasma mass spectrometry (ICP-MS), while biofilm formation and anti-bacterial response to free-floating cariogenic bacteria were investigated using anti-bacterial assays.

## 2. Materials and Methods

### 2.1. Sample Preparation

Cu-MBGN glass particles were prepared as mentioned previously in Zheng et al. [31]. The dental resin matrix was prepared by mixing Bisphenol A–glycidyl methacrylate (BisGMA; Merck, Darmstadt, Germany) and triethylene glycol dimethacrylate (TEGDMA; Merck, Darmstadt, Germany) in a 60/40 ratio and made photoactive by adding 0.2 wt% camphorquinone and 0.8 wt% ethyl-4-dimethylamino benzoate (Merck, Darmstadt, Germany). The resin was mixed with fillers to obtain eight experimental resin composites, the detailed composition of which is described in Table 1, while the composition of the fillers, as provided by the manufacturers, is listed in Table 2. The resin was preheated to 100 °C and fillers were gradually added under the orange illumination. Resin and fillers were mixed with an asymmetric centrifugal mixer (Speed Mixer 150 FVZ, Hauschild & Co. KG, Hamm, Germany) at 27,000 rpm for 5 min to obtain the composite materials.

### 2.2. Light Curing

Custom-made Teflon mold with 2 mm diameter and 3 mm height was filled with experimental resin composite. The experimental resin composites were placed in the mold using a stainless steel Heidemann spatula and round condenser, and trapped air was removed while applying gentle pressure. Excess material was also removed. Bluephase PowerCure (Ivoclar Vivadent, Schaan, Liechtenstein, 950 mW/cm^2^) was used to cure the experimental resin composite three times on each side for a total of six times and 120 s. Before removing from the mold, experimental resin composites were polished on both sides. They were then stored at room temperature in a sterile well plate.

### 2.3. Micro-Computed Tomography (Micro-CT) Imaging and Polymerization Shrinkage Analysis

To calculate the shrinkage caused by polymerization, samples were scanned both before and after light curing. Briefly, for each sample, a stopper was placed in the Kapton tube on top of the micro-CT specimen holder, and the experimental resin composite was transferred into a Kapton tube (2 mm diameter and 3 mm height) using a stainless steel Heidemann spatula and round condenser. Entrapped air was removed by pressing with light pressure, and the extra material was removed from the tube. The holder with the experiment sample was then transferred to the micro-CT and scanned using a desktop micro-CT scanner (Skyscan 1172, Bruker, Kontich, Belgium). The samples were scanned at 50 kV, 190 uA, at a pixel size of 3.9 µm and a rotation step of 0.79^o^ throughout the range of 360^o^. The total scanning time was around 12 min. Following the initial scan, the sample was cured three times on each side, for a total of six times and 120 s, without the sample being removed from the micro-CT machine. A second scan was performed using the same configurations and position. Finally, following the scanning, the samples were carefully removed from the Kapton tube, polished and kept in a sterile well plate to be further utilized for FTIR, porosity analysis and ion release. The same procedure was repeated for three samples from each group, and the data obtained were used to calculate polymerization shrinkage. Dragonfly software ORS ver. 3.6. (Object Research Systems Inc., Montreal, QC, Canada, 2022; software available at http://www.theobjects.com/dragonfly, accessed on May 2022) was used to calculate the shrinkage, volume of the entire cylinder of resin and to visualize the volumetric difference between pre- and post-curing of the composites.

### 2.4. Degree of Conversion

Attenuated total reflection Fourier transform infrared spectrometry (ATR-FTIR; Spectrum One, Stamford, CT, USA; PerkinElmer, Waltham, MA, USA) was used to analyze the percentage of reacted methacrylate groups (degree of conversion) of each material on the sample surface. Three measurements per group were performed (n = 3). The degree of conversion was calculated according to the “two-frequency baseline” method, where the percentage of unreacted methacrylate groups in the material is determined by using the ratio of peak intensities of aliphatic C=C mode at 1638 cm^−1^ and aromatic C···C band at 1608 cm^−1^ acquired from FTIR spectrum [40].
$$\mathrm{DC}\;\left(\%\right)=\left(\frac{\mathrm{aliphatic}\;C=C/\mathrm{aromatic}\;C=C\;\mathrm{polymer}}{\mathrm{aliphatic}\;C=C/\mathrm{aromatic}\;C=C\;\mathrm{polymer}}\right)\times 100\%$$

### 2.5. Nano-CT Imaging and Porosity Analysis

Nano-CT was performed to study the porosity of the samples at a sub-micrometer resolution. Following the shrinkage study, one cured sample from each group was randomly selected and scanned using a nano-CT system (Bruker Skyscan 2211, Bruker micro-CT, Kontich, Belgium). The images were acquired at a final isotropic resolution of 1.3 µm per pixel, at camera binning = 1 × 1, at 70 kV accelerating voltage, 210 μA current, and with a 0.125 mm aluminum filter placed in front of the camera. The samples were rotated 360° about their vertical axis using a step size of 0.37°, and the exposure time was set to 1100 ms per projection. An average of 4 frames was taken, the total being 4400 ms per projection. Images were reconstructed using NRecon software (Bruker, Kontich, Belgium) with a filtered back-projection algorithm with ring artefact correction of 9 and beam hardening correction of 60%. CTan (Bruker, Kontich, Belgium) was used to carry out the data analysis of the reconstructed 3D images. The pores were segmented, and porosity and pore morphometry were calculated from the segmented pores. For visualization, Dragonfly software ver 3.6 was used.

### 2.6. Ion Release Profile

Following micro-CT, we measured the ion release behavior of our scanned samples using an inductively coupled plasma mass spectrometer (ICP-MS, Bruker Aurora Elite quadrupole, Bruker Co., Fremont, CA, USA) equipped with a Cetac ASX-250 autosampler and an ESI OneFAST sample introduction system. Briefly, an immersion solution of 50 mmol/L 4-(2-hydroxyethyl)piperazine-1-ethanesulfonic acid, N-(2-hydroxyethyl)piperazine-N′-(2-ethanesulfonic acid) (HEPES; Sigma-Aldrich, St. Louis, MO, USA) and 133 mmol/L NaCl (VWR Chemicals, Darmstadt, Germany) was prepared in deionized water. Each sample was incubated at 37 °C on a plate shaker in the dark with 2 mL of immersion solution in a 15 mL propylene tube. For a total of 28 days, the samples were kept immersed. Fresh immersion solution was added when solutions from samples were collected on day 1, day 3, day 7, day 14 and day 28. In order to keep the supernatant for ICP-MS mass spectrometry analysis, 1% (*v*/*v*) single distilled nitric acid (HNO_3_) was added to each tube and kept at 4 °C. ICP-MS determined ionic concentrations of supernatant.

The calibration curve was calculated from a multielement calibration standard at 10, 100 and 1000 μg/L. In addition, an artificial drinking water standard (CRM-TMDW-A; High-Purity Standards, North Charleston, SC, USA) was run as an unknown. Concentrations of boron, calcium and copper were determined by analysis of masses 11, 44 and 65, respectively. Scandium (mass 45) was used as an internal standard for all measured elements such as boron (B), calcium (Ca), and copper (Cu), Table 3.

### 2.7. Surface Morphology and Particle Distribution of Fillers with SEM and EDX

After the ion release, samples were analyzed to study the composite matrix’s surface morphology and ion distribution. Samples were fractured, and one section from each sample was randomly selected to study the morphology by scanning electron microscopy (SEM) (Hitachi Analytical TableTop SEM TM3030, Hitachi, Tokyo, Japan). Elemental analysis for elemental distribution on the composites was performed by energy-dispersive X-ray spectroscopy (EDX) with the aforementioned apparatus. Elemental composition was recorded, and average values were calculated. The same setup was repeated for all the samples from the ion release test.

### 2.8. Biofilm Formation and Anti-Bacterial Activity

We studied the biofilm formation and anti-bacterial activity on the surface of the experimental resin composites using bacterial strains (*Streptococcus mutans* ATCC 35668 or *Actinomyces naeslundii* ATCC 19039, Manassas, VA, USA). Samples were placed in sterile Eppendorf tubes after being sterilized by UV (Hoefer UV Crosslinker, Hoefer, Inc., Holliston, MA, USA) at Energy 25 (2500 µJ/cm^2^) for 5 min on each side (10 min UV). *S. mutans* was cultured on brain heart infusion (BHI) agar at 37 °C and 5% CO_2_ for 24 h. Bacteria were inoculated in BHI medium and grown at 37 °C and 5% CO_2_ for 24 h. The overnight culture was diluted 100-fold in fresh BHI and grown at 37 °C and 5% CO_2_ to an optical density (OD600) of 0.6–0.7. *A. naeslundi* was cultured on BHI agar in an anaerobic atmosphere (5% H_2_, 5% CO_2_ and 90% N_2_) for 72 h before inoculation in 10 mL fresh BHI and growth to obtain OD600 = 0.6–0.7.

100 µL of bacterial suspension was added to each sample and incubated for 24 h at 37 °C and 5% CO_2_. The total number of viable bacteria in suspension (free-floating) and bacteria attached to the experimental resin composite (biofilm) were evaluated by counting the number of colony-forming units (CFU) (n = 3). Briefly, an aliquot of the bacterial suspension was removed, and the samples were washed quickly 3 times in sterilized distilled water and transferred to 1 mL fresh BHI. Bacteria attached to the samples were collected by dispersing the biofilm by vortexing for 1 min at 2400 rpm. Detached bacteria from biofilm and free-floating bacteria were serially diluted, spotted on the blood agar plates and incubated for 48 h at 37 °C and 5% CO_2_. Finally, colonies were manually counted.

### 2.9. Statistical Analysis

A Kolmogorov–Smirnov test was used to determine if the dataset’s distributions were parametric or nonparametric. After that, a normality test was performed (Holm-Sidak method). The data were given as arithmetic mean values with standard deviation when normally distributed. Regular ANOVA was used for the rest of the analysis, with a Tukey test for post-hoc comparison. All the bacterial datasets were tested with a two-tailed student *t*-test. GraphPad Prism 8 (GraphPad Software Company, San Diego, CA, USA) was used to conduct all analyses. The significance level was set to 0.05, 0.01 or 0.001.

## 3. Results

### 3.1. Characterization with SEM-EDX

Surface morphologies of binary experimental composites (Figure 1), ternary experimental composites (Figure 2), and control Si and BG composites were characterized using SEM. EDX analysis was used to quantitatively determine elements (Figure 3). Silica (Si), phosphorous (P), copper (Cu), zinc (Zn) and calcium (Ca) were mapped on the surface of the composite after 28 days of immersion in HEPES-buffered solution for the ion release test. Surfaces of experimental binary composite 10-CuBG and 10-BG and 10-Si composites (Figure 1) appeared rough. The SEM image of 10-BG showed a rough surface with clusters (Figure 1B—SEM). Our EDX data showed a homogenous distribution of elements on the surface of binary composites except for 10-BG, which showed the agglomerates of Ca (Figure 1B—EDX, Elemental). Similarly, SEM analysis revealed rough surfaces of experimental ternary composites and 15-Si (Figure 2A–E—SEM). 15-BG composite surface again appeared rough with clusters (Figure 2D—SEM). The elementals were homogeneously distributed on the surfaces of ternary composites (Figure 2A–C—EDX, Elemental, Figure 2E—EDX, Elemental) except 15-BG (Figure 2D—EDX, Elemental), where Ca was unevenly distributed and appeared to be present in the form of agglomerates all over the composite surface.

The atomic percentage of Si was above 92% in experimental composites, which was significantly higher in ternary composites than the 15-BG composite and was at the same level as Si composites (Figure 3A). The amount of P on the experimental composite appeared to depend on the Cu-MBGN amount. The expression of P on experimental composites decreased with an increasing amount of Cu-MBGN incorporation to 5.2% in 10% Cu-MBGN (10-CuBG-Si), 1.4% in 5% Cu-MBGN (5-CuBG-Si) and 0.6% in 1% Cu-MBGN (1-CuBG-Si) (Figure 3B). However, P expression in experimental composites was significantly lower than in BG composites. In addition, 0.6% of P was identified on the surface of Si composites.

On the other hand, the amount of Cu identified on the surface of experimental composites was directly proportional to the percentage of Cu-MBGN incorporation (Figure 3C). It was significantly higher in 10% Cu-MBGN experimental composites in binary (2.25%) and ternary (2.27%) fillers. Its value was reduced to 0.94% in the experimental composite with 5% Cu-MBGN and 0.33% in 1% Cu-MBGN experimental composite, which was similar to Si and BG composites (0.4%). There was no statistical difference in Ca expression between the different composites (Figure 3D).

### 3.2. Polymerization Shrinkage using Micro-CT and Nano-CT

We studied the volumetric polymerization shrinkage in experimental resin composites using micro-CT. The results (Figure 4) showed that polymerization shrinkage after light curing in binary experimental filler (10-CuBG) was within the range of 10-BG and 10-Si, 4.22%, 4.31% and 4.24%, respectively. In addition, polymerization shrinkage in binary fillers appeared to be higher than in ternary fillers. There was no prominent dose-dependent effect on polymerization shrinkage in experimental ternary fillers. 5% Cu-MBGN and 10% Cu-MBGN incorporation in the experimental composite gave a shrinkage of 4.00% and 4.05%, respectively; 1% Cu-MBGN incorporation in the experimental composite caused 3.91% shrinkage. Polymerization shrinkage was significant between 15-BG and 15-Si. In general, the polymerization shrinkage in ternary experimental composites was significantly higher than in control composites 15-Si and 15-BG.

In order to assess more detailed descriptions on where in the composite the shrinkage took place, composites where scanned pre- and post-curing without removal from the nano-CT (Figure 5). The generated 3D models for the pre- and post-cured samples could be overlayed, allowing for an exact superimposure. The advantage is that one can visualize where the shrinkage was highest and where it was lowest. From both binary and ternary composites, inert and experimental composites with 10% Cu-MBGN were chosen to demonstrate the visual representations of polymerization shrinkage (Figure 6). A very thin area of shrinkage, represented by the color red, was seen after polymerization around the peripheral margins of the composites (Figure 6A,E,I,J,M), as well as inside the composites, as seen in the cross-sectional images (Figure 6B,F,J,M). For the simpler presentation of polymerization shrinkage, the cross-sectional images (Figure 6C,D,G,H,K,L,O,P) from nano-CT data were created in which the red color presents the dimensions of composites before curing and the blue color presents the dimensions of composites after curing. Finally, the polymerization shrinkage was observed along the margins of the composites and within the pores inside the composites, as shown in Figure 6C,D,G,H,K,L,O,P.

### 3.3. Porosity by Nano-CT

Total porosity results (Figure 7A) indicated that 10-Si and 10-BG had the highest (6.99%) and lowest (0.51%) percentages of total porosity compared to the other groups, respectively. Cu-MBGN incorporation in binary filler (10-CuBG) significantly reduced the porosity (3.9%) compared to 10-Si. In the ternary filler group, 15-BG and 15-Si showed total porosity of 4.15% and 3.82%, respectively. 1% Cu (1-CuBG-Si) increased the total porosity to 4.42% compared to 15-BG and 15-Si. However, total porosity was significantly reduced with an increase in the amount of Cu-MBGN: 5% Cu-MBGN (5-CuBG-Si) had a total porosity of 2.12%, and 10% Cu-MBGN (10-CuBG-Si) had a total porosity of 1.19%.

We observed a high percentage of open porosity (Figure 7B) in 15-BG composite (3.6%) compared to experimental composites and inert composites. In binary composites, the 10-CuBG showed significantly higher open porosity (1.3%) than 10-BG (0.19%) and significantly lower compared to 10-Si (1.7%). On the other hand, 10% and 1% Cu-MBGN incorporation in ternary composites (10-CuBG-Si and 1-CuBG-Si) decreased the open porosity to 0.16% and 0.08%, respectively. In comparison, 5% Cu-MBGN incorporation (5-CuBG-Si) increased the open porosity (1.05%).

In contrast, the percentage of closed porosity (Figure 7D) appeared to be significantly highest in 10-Si (5.2%) and lowest in 10-BG (0.31%). The incorporation of Cu-MBGN in binary filler (10-CuBG) significantly reduced the closed porosity to 2.5%. 1% of Cu-MBGN addition (1-Cu-BG-Si) in ternary filler drastically increased the percentage of closed porosity (4.3%), which was higher than 15-Si (2.7%) and 15-BG (0.57%). On the other hand, an increased amount of Cu-MBGN reduced the percentage of closed porosity and amounted to 1.07% for 5-Cu-BG-Si and 1.04% for 10-Cu-BG-Si. The percentage of closed porosity was lowest in 15-BG (0.57%). Ternary filler containing 1% Cu-MBGN (1-Cu-BG-Si) showed the highest object surface/volume ratio compared to the other groups (Figure 7C).

### 3.4. Degree of Conversion

The degree of conversion of all the composite samples was over 70%, except 10-CuBG (63%) and 10-CuBG-Si (56%) (Figure 8). Both 10-BG and 15-BG showed a very high degree of conversion above 90% and were significantly higher than the other materials. An increase in the amount of Cu-MBGN in the ternary filler reduced the degree of conversion from 75% in 1-CuBG-Si to 71% in 5-CuBG-Si, and 56% in 10-CuBG-Si. Similarly, adding Cu-MBGN into binary filler significantly reduced (63%) the degree of conversion of 10-CuBG compared to control binary materials.

### 3.5. Ion Release Profile with ICP-MS

Ion concentration curves demonstrating the ion release profile at each time point and cumulative mass release of B, Ca and Cu during the observation period of 28 days are shown in Figure 9. 15-BG showed the highest amount of B and Ca release (Figure 9A,B) compared to experimental composites containing Cu-MBGN. Following the release of B in 15-BG, a drastic increase in B release was observed on day 3, which increased further on day 7, followed by a decrease on day 14 and an increase on day 28. An initial burst of Ca on day 1 was observed for the same material, followed by a drastic decrease on day 3. A gradual increase followed this in releasing Ca concentrations during the observational period of 28 days. A similar pattern of Ca release was demonstrated by 10-BG, except on day 14 and day 28 where Ca ion release remained constant. In 15-BG the concentration of B release remained constant on day 1 and day 3, followed by a marked and continuous increase.

The release of B from the 10-Si and 15-Si composites continuously increased during the 28 days. However, it was higher in 15-Si. The concentrations of Ca and Cu (Figure 9B,E) from the 10-Si and 15-Si composites were negligible compared to the experimental composite.

The release of B and Ca ion concentration was lower in experimental composites compared to 15-BG, but no dose-dependent effect on release was observed. However, the trend of B ion release was similar in the experimental groups: initial high release followed by a minor decrease and then a gradual increase over a defined period. The concentration of B remained the lowest in 10-CuBG compared to other experimental composites. Experimental composites with a low Cu-MBGN concentration (1-CuBG-Si and 5-CuBG-Si) showed the same pattern of Ca release; Ca ions were released in a burst on day 1 (higher in 5-CuBG-Si), but Ca ions release drastically dropped on day 3, followed by a gradual decrease. Whereas in experimental composites with a high concentration of Cu-MBGN (10-CuBG and 10-CuBG-Si), Ca ions were released in a burst on day 1, but their release decreased drastically on day 3, followed by a gradual increase on the respective time points.

In contrast, the Cu ions concentrations for experimental composites were superior to those of the BG and Si-composites (Figure 9E). A dose-dependent increase in Cu ions release concentration was identified, which was high in composites with the highest amount of Cu-MBGN incorporation (10%) (10-CuBG-Si and 10-CuBG) and decreased with the lower amount of Cu-MBGN incorporation in experimental composites (5-CuBG-Si and 1-CuBG-Si respectively). The pattern of Cu release was similar in composites with high Cu-MBGN incorporation: an initial burst of Cu ions release followed by a sudden decrease on day 3, then a gradual reduction until day 14, followed by an increase on day 28. Following the initial burst on day 1, experimental groups with lower Cu incorporation showed a gradual decrease in Cu ion release in composites with 5% Cu-MBGN (5-CuBG-Si). In contrast, it remained constant in composites with 1% Cu-MBGN (1-CuBG-Si). The cumulative concentration of B, Ca, and Cu ion release (Figure 9B,D,F) demonstrated a similar ion release profile as mentioned above. The cumulative release mass of B and Ca in 15-BG remained relatively superior and increased continuously to those of experimental composites. On the other hand, B release concentration by other groups remained within the range of experimental composites (Figure 9B). The concentration of released Ca remained lowest in 10-Si and 15-Si compared to the experimental groups (Figure 9D). The cumulative Cu concentration release plots indicated a dose-dependent increase in Cu for experimental groups. The lowest release of Cu concentration was observed in 1% Cu-MBGN incorporated experimental composite and was marginally higher than Si and BG composites. In contrast, high initial bursts and continuous gradual increases in the release of Cu were observed in 10%, both ternary and binary fillers and 5% Cu incorporated experimental groups.

### 3.6. Anti-Bacterial Effect and Biofilm Formation on the Surface of the Composite Specimens

We investigated the anti-bacterial effect of the composites and biofilm formation by *Streptococcus mutans* (*S. mutans*) and *Actinomyces naeslundii* (*A. naeslundii*) (Figure 10). The anti-bacterial effect of the experimental composites in the binary and ternary groups on *S. mutans* was stronger than that of BG and Si composites (Figure 10A). 10-CuBG showed a 0.25-fold lower amount of free-floating *S. mutans* than 10-Si. Ternary experimental composites had a dose-dependent effect on *S. mutans*: increasing the amount of Cu-MBGN in the composite decreased the amount of free-floating *S. mutans*: 1-CuBG-Si showed a 0.45- and 0.11-fold decrease in S. mutans compared to 15-BG and 15-Si, respectively, while 5-CuBG-Si and 10-CuBG-Si showed a 0.58- and 0.77-fold decrease in *S. mutans* compared to 1-CuBG-Si, respectively. The amount of free-floating *S. mutans* was significantly lower in 10-CuBG-Si (*p* < 0.01) than in 15-Si.

On the other hand, biofilm formation in both the binary and ternary groups was observed to be higher in BG composites than in other groups (Figure 10B). The amount of biofilm formed on 10-BG composites was 0.99-fold higher than for 10-Si, whereas biofilm formation on 15-BG was 2.56-fold higher than 15-Si. However, the differences were not statistically significant. Biofilm formation in 10-CuBG appeared to be 1.44- and 0.45-fold less than 10-BG and 10-Si but the differences were not statistically significant. However, among the experimental composites in the ternary groups, 5-CuBG-Si exhibited the highest amount of bacterial attachment (0.79-fold) compared to other experimental groups. 1-CuBG-Si and 10-CuBG-Si showed 0.09- and 0.32-fold decreases in *S. mutans* attachment, respectively, compared to 5-CuBG-Si. The attachment of *S. mutans* was significantly lower in 10-CuBG-Si compared to 1-CuBG-Si (*p* < 0.05).

In a bacterial test with *A. naeslundii* (Figure 10C,D), the experimental composites with 10% Cu-MBGN incorporation showed a reduction in both free-floating bacteria (0.11-fold in 10-CuBG and 0.08-fold in 10-CuBG-Si) and biofilm formation (0.14-fold in 10-CuBG and 0.63-fold in 10-CuBG-Si) compared to 10-Si and 15-Si, respectively. The experimental composite 5-CuBG-Si showed a 0.15-fold reduction in free-floating bacteria compared to 15-Si (0.92) and a 0.26-fold decrease in bacterial attachment compared to 15-Si (1.48). Similarly, 1-CuBG-Si showed a 0.03-fold increase in free-floating bacteria compared to 15-Si (0.92), and a 0.26-fold decrease in biofilm formation compared to 15-Si (1.48). In addition, free-floating bacteria and bacterial attachment to 15-BG were observed to be lower compared to 15-Si.

## 4. Discussion

In this study, Cu-MBGN, an antimicrobial and remineralizing component, was incorporated into a resin matrix to develop experimental dental composites that could exert anti-bacterial activity and prevent secondary caries. Our results show that the Cu-MBGN-containing materials exhibit better antimicrobial activity against *S. mutans* than composites with conventional 45S5. The Cu-MBGN materials showed a dose-dependent Cu release, with the release being highest on the first day, followed by a further increase after 28 days. Accordingly, 10% Cu-MBGN had the best anti-bacterial effect, as evidenced by the lowest attachment of free-floating bacteria and the lowest biofilm formation. In contrast, 45S5-containing materials showed the highest adhesion of *S. mutans*. Ca release was highest in the bioactive control with 15% 45S5 BG, which correlated with the highest number of open porosities on the surface. Polymerization shrinkage reached similar values for all tested materials. Surprisingly, it did not show the usually observed correlation with the degree of conversion.

Polymerization shrinkage and its precise measurement remain a challenge to the development of compatible restorative materials [41,42]. Previously, dilatomers and scales where used to measure composite shrinkage [43], until Sun et al., presented a method using micro-CT to measure the polymerization shrinkage [44]. The drawback of method by Sun et al. is that it does not allow for exact superimposure of the pre- and post-cured samples, as there are too few landmarks to match the two different 3D models. Here, we present a new method that allows for such a superimposure. This technique allows for both visualization and quantification of spatial shrinkage. With the development of this technique, we conducted a systematic investigation to determine the pre- and post-polymerization volumetric change and successively measured the shrinkage. However, to increase the precision of the shrinkage measurements, composites in this study were cured in the same location in the nano-CT instrument to avoid changes in the reorientation of the samples before and after curing. Therefore, it can be claimed that this investigation precisely validates the polymerization shrinkage in experimental composites.

### 4.1. Binary Filler Group

A binary filler group was formed to allow direct comparison of materials with identical resin and filler base (barium glass microfiller) and the same 10% filler amount of either inert (silica nanofiller, 10-Si), bioactive control (45S5 BG microfiller, 10-BG) or experimental fillers (Cu-MBGN, 10-CuBG).

SEM-EDX showed a homogeneous distribution of the fillers for all materials of the binary filler group. Well-blended and uniformly distributed fillers are a prerequisite for any experimental composite, especially if nanoscale fillers are included in the composition. Nanofillers increase the viscosity of the composite and severely limit the maximum filler amount due to the large surface area. The mesoporous particles used in this study have a much larger diameter than the silica nanofillers (100 nm and 12 nm, respectively). However, the surface area of Cu-MBGN reaches almost twice that of silica nanofillers due to surface porosity [31], making it even harder to mix than the nanoparticles. Consequently, 10-Si (and 10-BG) material had a flowable consistency, while 10-CuBG had a higher viscosity at the same resin amount.

Agglomerates visible in 10-BG are the only non-uniform surface morphology. Agglomerates of 5–10 µm (Figure 1B) with high Ca and P content is easily recognized as 45S5 glass particles. The manufacturer’s data show that their d_50_ diameter is 4 µm but can reach 13 µm (d_99_). Phosphorus also predominates in 10-BG compared to the other two materials in the binary filler group, as 6% P_2_O_5_ is present in the structure of 45S5 BG. Despite the absence of P in 10-CuBG and 10-Si, traces of P were most likely observed due to immersion in the HEPES-buffered solution. Similarly, 10-CuBG has the highest amount of Cu, which should not be present in 10-BG and 10-Si. However, it is detected by EDX and in the ion release study with a highly sensitive ICP-MS instrument. A possible origin of Cu in non-Cu MBGN materials is the stainless steel instruments used to make the composite samples.

When immersed in simulated body fluid (SBF), free Cu-MBGN particles release most Cu and Ca ions in the first 24 h [31]. This behavior is typical of mesoporous particles prepared by the sol–gel process, as they have a large surface area and high reactivity. In the present study, a similar burst from the 10-CuBG composite was observed on day 1 in the HEPES-buffered solution, but the Cu and Ca concentrations decreased continuously until day 14, followed by an increase on day 28. While direct contact of the entire surface of the Cu-MBGN particles with SBF resulted in rapid and complete depletion of ions, incorporating Cu-MBGN into a hydrophobic resin matrix provided sustained ion release. Initial contact of water with the Cu-MBGN particles exposed on the surface of the composite sample caused a sharp Ca and Cu increase on the first day. However, all composites exhibit different water sorption and solubility degrees depending on their composition and the experimental conditions [45,46]. The water penetrates through irregularities and porosities on the material’s surface and continues its path through the free space to the center of the sample. In this study, we used a conventional BisGMA/TEGDMA resin base. BisGMA contains OH groups and TEGDMA contains hydrophilic ester bonds that facilitate water sorption [47]. Water sorption is enhanced when bioactive substances such as ACP or BG [30] are added. In our study, water sorption and solubility were not evaluated, but higher sorption than for commercial composites is anticipated. Ion release is expected to be delayed by the time required for water to reach the Cu-MBGN particles in the center of the sample.

This study shows the secondary jump in Ca and Cu ion release from 10-CuBG at 28 days. In a clinical setting, this property could be advantageous as the initial Cu release could immediately act against bacteria remaining in the tooth cavity after preparation and later intercept bacterial adhesion and biofilm formation after restoration placement, thus preventing secondary caries [48]. Confirmation of this theory can be seen in the antimicrobial tests. The significantly highest Cu release in 10-CuBG from the binary filler group was responsible for the lower number of free-floating CFU and biofilm formation for both *S. mutans* and *A. naeslundii*. Similar to 10-CuBG, a reduction in free-floating bacteria was also observed with 10-BG, but a 0.99-fold higher amount of biofilm formed on the surface of 10-BG than on the inert control 10-Si. The anti-bacterial effect of 10-BG derives from a conventional 45S5 BG, which causes alkalinity of the immersion medium resulting from the release of OH ions from 45S5 BG [49,50,51,52]. In contrast, the antimicrobial effect of Cu is due to direct contact with bacteria or viruses, which is a probable reason for lower biofilm formation. Cu alters the permeability of bacterial cell membranes and accumulates in the cell, changing biochemical processes by interacting with cell enzymes and DNA and causing cell death [53]. Information on the adhesion of bacteria to the surface of a composite and biofilm formation is more meaningful information for the prevention of secondary caries. *S. mutans* is considered a successful colonizer and a highly competitive and prosperous bacterium in dental biofilms on caries-affected dental hard tissue surfaces. Its acidogenicity and acidurance are most likely responsible for *S. mutans* being one of the main bacterial species responsible for the occurrence of dental caries [54,55]. *A. naeslundii*, on the other hand, is a pioneer bacteria in oral biofilm formation and has been frequently found in cervical caries, but is less dominant in the microbiome of mature dental plaque [55,56]. A multispecies biofilm accounting for bacterial interactions would provide the most information on the prevention of secondary caries and is planned as the next step in this evaluation [57,58,59]. Still, despite the somewhat weaker anti-bacterial effect on *A. naeslundii* than a strong antimicrobial action on *S. mutans,* the preventive effect of 10-CuBG on secondary caries is still adequate.

Volumetric polymerization shrinkage was in a narrow range of 3.8% to 4.2% for all the composites tested in this study, slightly higher than for sculptable commercial composites (2% to 3%) and lower than values reported for flowable composites (5%) [60]. Although several statistically significant effects were found, the largest difference was between the binary and ternary filler groups. The values for all three materials averaged 4.2% in the binary group, ranging from 3.8 to 4% in the ternary group. The results are probably related to the resin content, which was higher in the binary group (35 vs. 30 wt%). Braga et al. described that the volumetric shrinkage is influenced by the filler volume fraction and the degree of conversion [60]. This result is expected since more resin is available for shrinkage in binary composites.

### 4.2. Ternary Filler Group

The ternary filler group investigated the dose-dependent effect of the combined action of 1, 5 or 10 wt% Cu-MBGN and silanized silica up to 15% (1-CuBG-Si, 5-CuBG-Si and 10-CuBG-Si, respectively) with the Ba glass microfiller base. This group was also included in the bioactive control with 15 wt% of 45S5 BG (15-BG) and the inert control with 15 wt% silica nanofillers.

Like in the binary filler group, SEM inspection of the micromorphology of the ternary group also showed a uniform distribution of fillers, revealing a denser structure on the 10-CuBG-Si than on 1-CuBG-Si or 15-Si. Due to the largest share of Cu-MBGN particles and silica nanoparticles, this was also macroscopically the most viscous material that required strong condensing. Consequently, this composite was one of the least porous materials in this study. The percentage of total porosity decreased with the increasing amount of Cu-MBGN and thus increased the CuBG composites’ viscosity. The porosity is one of the rarely investigated material properties strongly influencing composite materials’ physical and mechanical properties, affecting the crack initiation and directing the crack propagation upon exposure to high load [61]. The lack of porosity could explain the high flexural strength of 10-CuBG-Si in a previous study [38].

Regarding ion release, increased porosity enhances ion release from bioactive components in composite materials [62]. In contrast, a smooth, superficial, resin-rich layer prevents water uptake of the bioactive fillers and their solubility. The presence of porosity, while deteriorating the mechanical properties, increases the surface area exposed to water and improves water penetration; this increases the release of components with possible remineralizing and antimicrobial activity. Open pores on the material surface appear to have a greater effect on ion release than closed pores inside the material. In this study, material 15-BG exhibited the highest Ca release, which was the highest on the first day. At the same time, 15-BG also had the highest percentage of open pores, which could be partially responsible for the strong release on the first day. The main factor for the highest Ca release in 15-BG is the 45S5 BG, which contains 2.5 times more CaO than Cu-MBGN. In addition, the sodium contained in 45S5 is known to be a very hygroscopic element, resulting in higher solubility of 45S5 [50]. However, the dissolution of 45S5 BG from the surface of 15-BG (and 10-BG) must have resulted in increased water accumulation and decomposition of the Ba-glass microfillers (barium boro-alumino-silicate glass) used as base fillers in all materials. The degradation of the inert barium boro-alumino-silicate glass is evident from the increased B release, which was the only possible B source. Although B was detected in the immersion solution of all tested materials, the B concentration in 10-BG and 15-BG was 2–3 times higher than in the experimental composites or the inert controls. Moreover, the concentration increased at the end of the measurement period, indicating progressive material degradation. This result is consistent with the previously reported reduction in flexural properties of 45S5 BG-containing materials [38].

The anti-bacterial effect of Cu on *S. mutans* is also confirmed for the experimental CuBG composites. The dose-dependent Cu concentrations in the immersion solutions were inversely related to the number of free-floating *S. mutans*. The formation of monospecies biofilms on the surface of the CuBG composites was also reduced. On the other hand, a higher number of *S. mutans* CFU was found on the surface of 15-BG. The increased percentage of open pores in 15-BG probably increased the surface roughness of the material, which might have facilitated bacterial attachment [63,64]. Anti-bacterial properties have been reported for 45S5 [65]. They are mainly related to the increase in pH, which is a hostile environment for most bacteria.

The current composition of CuBG composites apparently did not result in the desired reduction in polymerization shrinkage, despite the reduction in the degree of conversion. However, the degree of conversion decreased with the increase of the Cu-MBGN content in the ternary filler group. This decrease is probably related to the high viscosity already mentioned. The increased surface area of the mesoporous particles and the presumed presence of supraparticles in the 10-CuBG-Si could be responsible for light attenuation and lower light transmittance [39]. Consequently, there is a lower number of activated sites for polymerization initiation. At increased viscosity and high filler volume, the mobility of the reactive species is limited, which probably leads to a lower degree of conversion. Similar results were reported in a previous study, although the reduction effect was much lower due to the higher amount of resin in the composition [39]. In contrast to the results of this study, the polymerization shrinkage was reduced with increasing Cu-MBGN content. However, there were several different circumstances in the previous study: linear shrinkage was measured, unlike the volumetric shrinkage measurement; the resin amount was previously higher, and the radiant exposure was five times lower. Therefore, additional fine-tuning of the resin content is required to achieve optimum polymerization properties.

## 5. Conclusions

This study showed that the anti-bacterial effect of copper-doped mesoporous bioactive glass nanospheres was maintained when they were embedded in a dental resin matrix with other inert fillers. The release of calcium and copper ions was sustained and detectable until the end of the measurement (28 days). The experimental composites had an anti-bacterial effect against free-floating *S. mutans* and significantly reduced the formation of a *S. mutans* biofilm. Further fine-tuning of the resin matrix and resin/filler ratio is needed to achieve lower polymerization shrinkage and higher conversion efficiency. Considering the results of our previous studies, we could conclude that the material with the combination of 5 wt% copper-doped mesoporous bioactive glass and silica nanofillers has appropriate mechanical properties, polymerization efficiency, and ion-releasing and anti-bacterial properties.

## Figures and Tables

**Figure 1 pharmaceutics-14-02241-f001:**
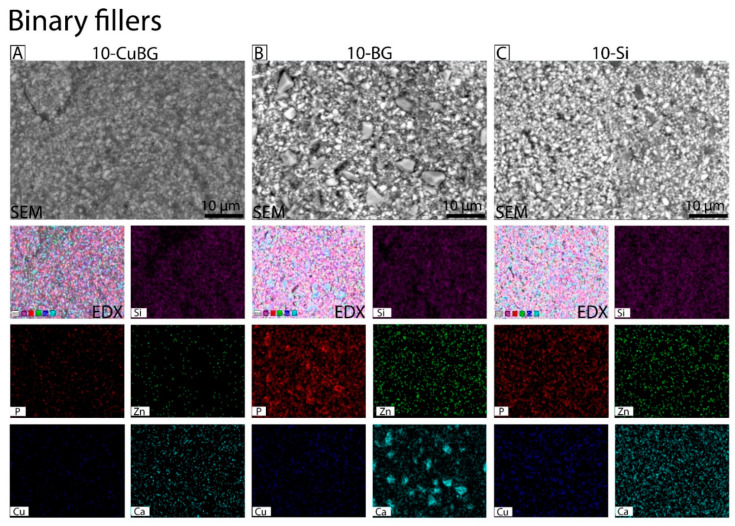
Representative surface morphology and elemental mapping images of the Binary filler group by SEM-EDX. (**A**) 10-CuBG, (**B**) 10-BG, and (**C**) 10-Si. SEM-EDX images were taken after 28 days of experimental composites immersed in HEPES buffer (pH = 6.5), and the elements identified were silica (Si) in purple, phosphorous (P) in red, zinc (Zn) in green, copper (Cu) in blue and calcium (Ca) in light blue. Scale bar corresponds to 10 µm in all micrographs.

**Figure 2 pharmaceutics-14-02241-f002:**
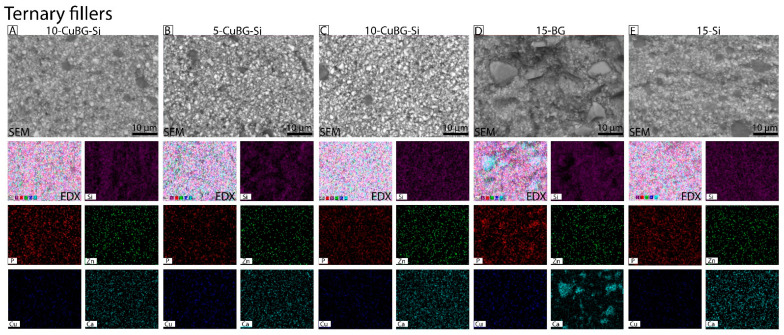
Representative surface morphology and elemental mapping images of the Ternary filler group by SEM-EDX. (**A**) 1-CuBG-Si, (**B**) 5-CuBG-Si, (**C**) 10-CuBG-Si, (**D**) 15-BG, and (**E**) 15-Si. SEM-EDX images were taken after 28 days of experimental composites immersed in HEPES buffer (pH = 6.5), and the elements identified were silica (Si) in purple, phosphorous (P) in red, zinc (Zn) in green, copper (Cu) in blue, and calcium (Ca) in light blue. The scale bar corresponds to 10 µm in all micrographs.

**Figure 3 pharmaceutics-14-02241-f003:**
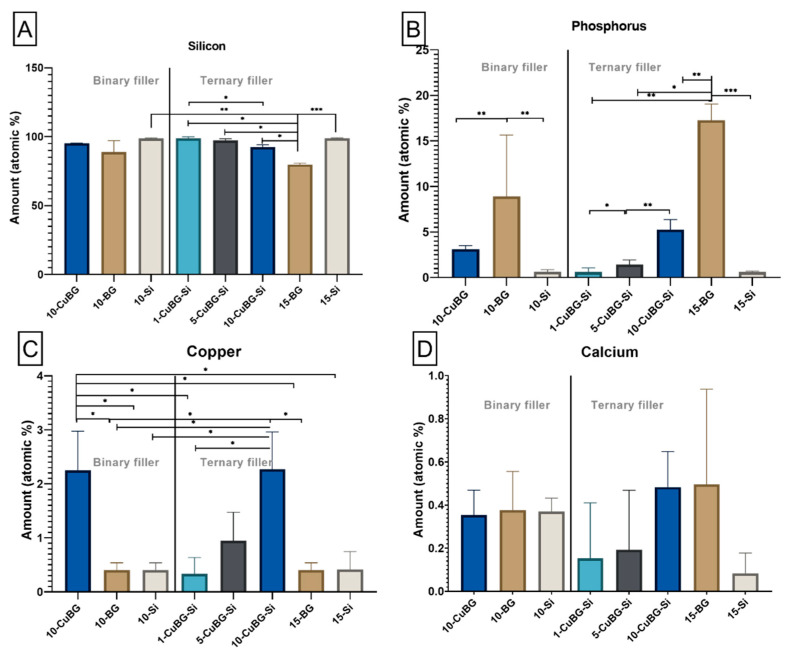
Quantitative analysis of elements is expressed in Figure 1 and Figure 2. The atomic percentage of (**A**) silicon, (**B**) phosphorous, (**C**) copper, and (**D**) calcium. Data presented as mean values with standard deviations. An asterisk indicates statistically significant differences (* *p* < 0.05, ** *p* < 0.01, *** *p* < 0.001).

**Figure 4 pharmaceutics-14-02241-f004:**
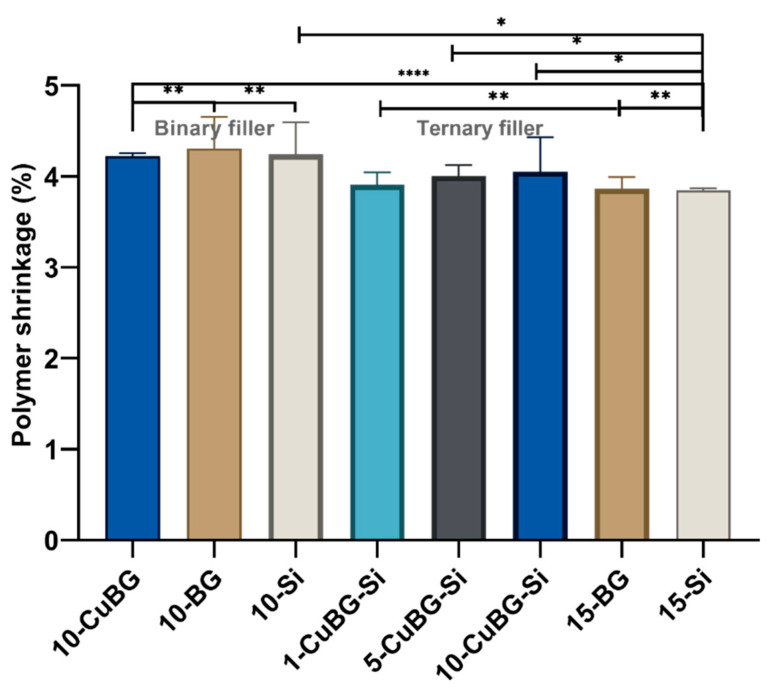
Polymerization shrinkage of the Binary filler and Ternary filler groups. Experimental composites were scanned using micro-CT before and after curing. Error bars represent standard deviations. A total of three samples were used per group (n = 3). Data presented as mean values with standard deviations. An asterisk indicates statistically significant differences (* *p* < 0.05, ** *p* < 0.01, **** *p* < 0.001).

**Figure 5 pharmaceutics-14-02241-f005:**
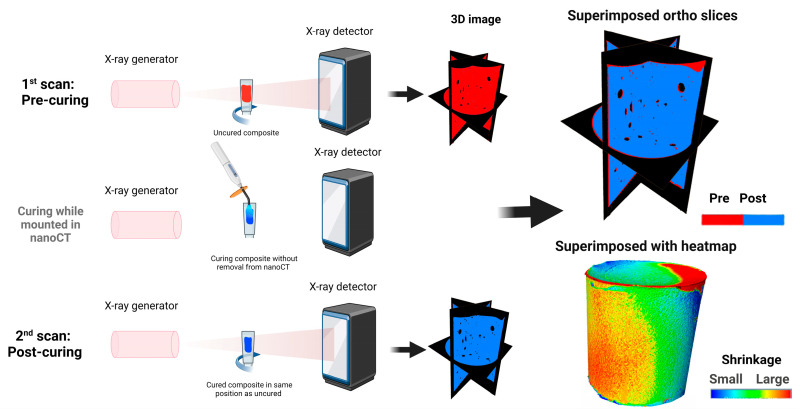
Schematic drawing on how the 3D shrinkage was assessed with nano-CT. The uncured composite was scanned, and a 3D model of the pre-curing was made. The same composite was cured inside the nano-CT with the same lamp and exposure time as described in the Materials and Methods section. The cured composite was scanned again. Since the composite was never removed from the nano-CT, it is possible to superimpose the pre- and post-curing, allowing for exact quantification and visualization of where the shrinkage occurred.

**Figure 6 pharmaceutics-14-02241-f006:**
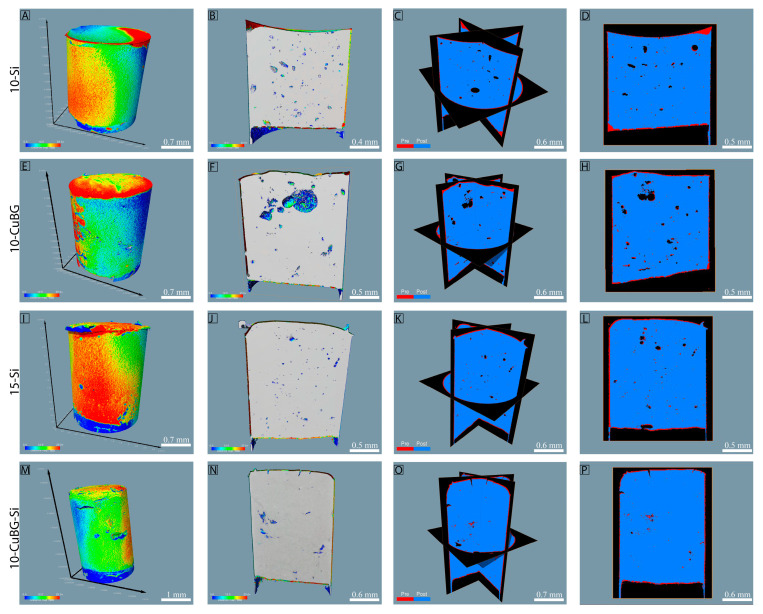
Superimposed nano-CT images and reconstruction of experimental composites containing binary and ternary fillers. (**A**,**E**,**I**,**M**) shows composite specimens with overlaid dimensional changes before and after polymerization, labeled nano-CT image overview. (**B**,**F**,**J**,**N**) represents the cross-sectional images, showing the pre- and post-polymerization areas of composites given in (**A**,**E**,**I**,**M**). The cross-sectional images of composites (given in (**A**,**E**,**I**,**M**)) that show the dimensions of the samples before (red) and after (blue) polymerization are represented in (**C**,**D**,**G**,**H**,**K**,**L**,**O**,**P**). The presented images are superimposed images of before and after curing experimental composites.

**Figure 7 pharmaceutics-14-02241-f007:**
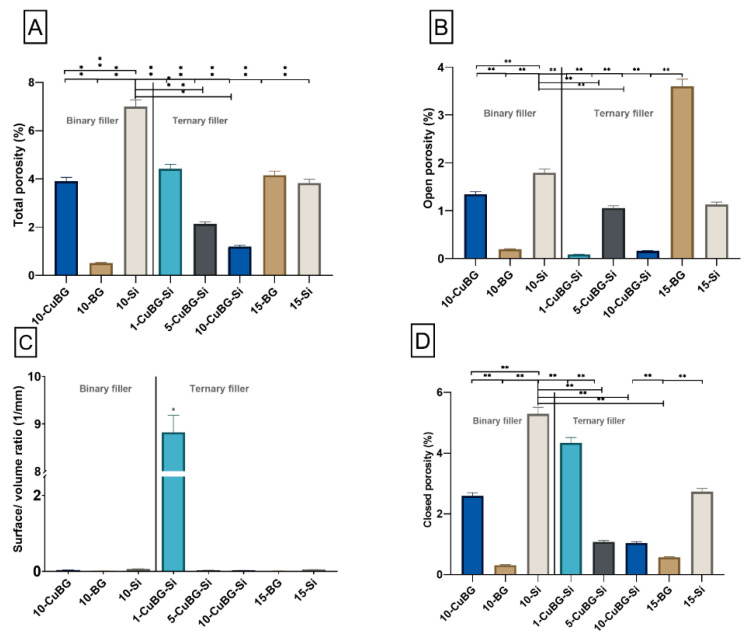
Graphical illustration of qualitative analysis of porosity of materials in the Binary filler and Ternary filler groups. (**A**) Total porosity of experimental composites, (**B**) percentage of open porosity in experimental composites, (**C**) surface and volume ratio of composite and (**D**) percentage of closed porosity in experimental composites. (**A**–**D**) were analyzed and calculated by using nano-CT and Amira-Avizo 3D Software ver 2022.1. Three samples from each group (n = 3) were used for analysis. Data presented as mean values with standard deviations. An asterisk indicates statistically significant differences (* *p* < 0.05,** *p* < 0.01).

**Figure 8 pharmaceutics-14-02241-f008:**
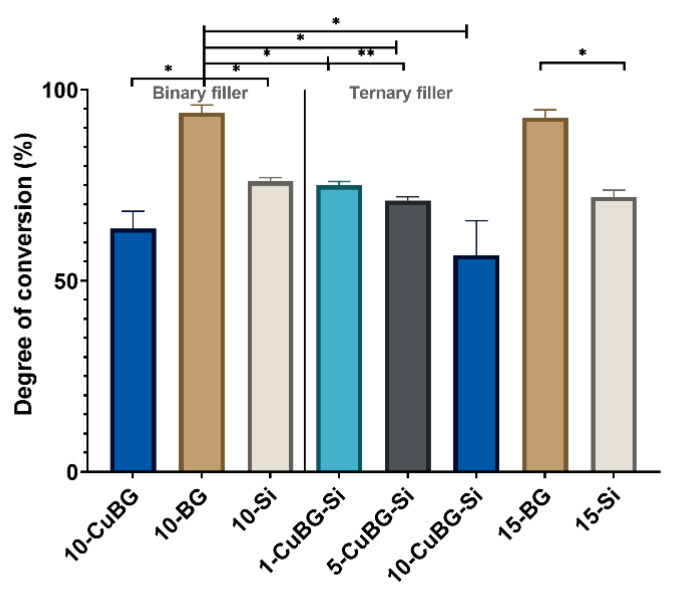
Degree of conversion of the Binary filler and the Ternary filler group after polymerization. Three samples from each group (n = 3) were used for analysis. Error bars represent standard deviations. An asterisk indicates statistically significant differences (* *p* < 0.01, ** *p* < 0.001).

**Figure 9 pharmaceutics-14-02241-f009:**
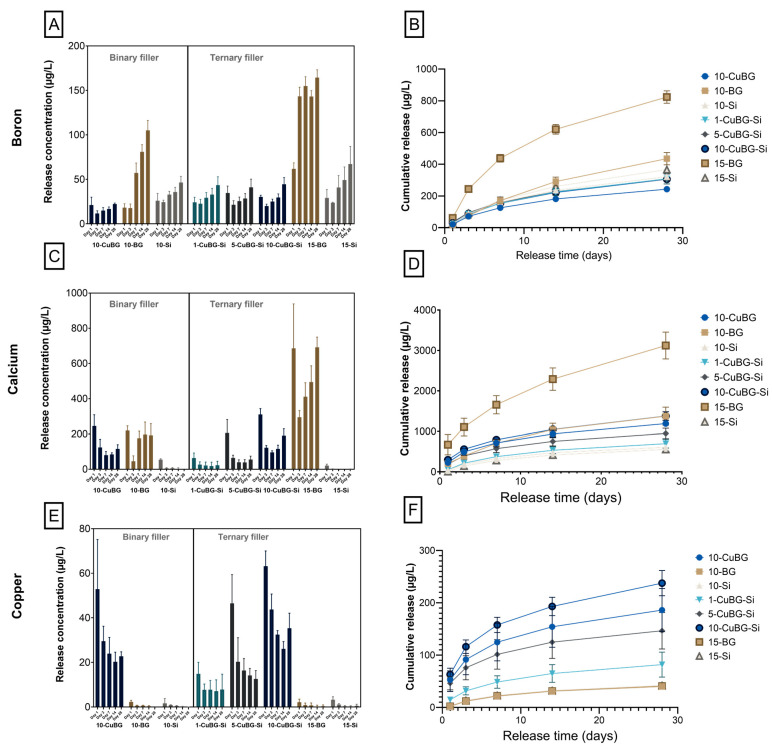
The ion release profile of the Binary filler and Ternary filler groups over 28 days. (**A**) The concentration of boron ion released, (**B**) cumulative concentration of boron ion released, (**C**,**D**), concentration of calcium ion released, and (**E**,**F**) concentration of copper ion released. All the samples were immersed in HEPES buffer (pH = 6.5). Three samples from each group (n = 3) were used for analysis. Data presented as mean values with error bars representing standard deviations (significance levels not shown for display reasons).

**Figure 10 pharmaceutics-14-02241-f010:**
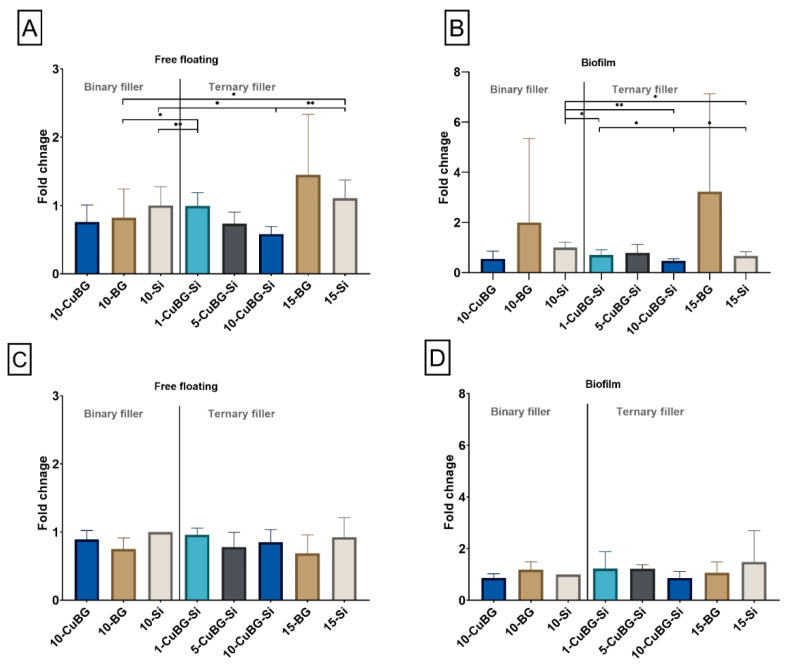
Growth and biofilm formation of *Streptococcus mutans* and *Actinomyces naeslundii*. (**A**) Relative amount of free-floating bacteria in a culture of *S. mutans* after 24 h of incubation in the presence of the respective composites, (**B**) relative amount of biofilm formation by *S. mutans* on composite surfaces after 24 h of incubation, (**C**) relative amount of free floating *A. naeslundii* after 24 h of incubation in the presence of composites, and (**D**) relative amount of biofilm formation by *A. naeslundii* on composites surface after 24 h of incubation. Three samples from each group (n = 3) were used for analysis. Data presented as mean values with standard deviations. (* *p* < 0.05, ** *p* < 0.01).

**Table 1 pharmaceutics-14-02241-t001:** Composition of experimental resin composites. Resin is 60/40 BisGMA/TEGDMA. All the amounts are expressed in wt%.

Group	Material	Resin	Ba Glass Microfillers	Cu-MBGN	Silica Nanofillers	45S5 BG
**Binary filler group** **(65% filler load)**	10-CuBG	35%	55%	10%	-	-
	10-BG			-	-	10%
	10-Si			-	10%	-
**Ternary filler group** **(70% filler load)**	1-CuBG-Si	30%	55%	1%	14%	-
	5-CuBG-Si			5%	10%	-
	10-CuBG-Si			10%	5%	-
	15-BG			-	-	15%
	15-Si			-	15%	-

**Table 2 pharmaceutics-14-02241-t002:** Characteristics of fillers used in the present study (data provided by the manufacturers).

Name	Type	Manufacturer/ Product	Composition (wt%)	Size (d50)	Silanization
Cu-MBGN	Experimental/Bioactive	Produced in-house [31]	SiO_2_ 84.8% CaO 9.4% CuO 5.8% *	~100 nm	No
45S5 BG	Commercial/Bioactive	Schott, Mainz, Germany G018-144	SiO_2_ 45% Na_2_O 24.5% CaO 24.5% P_2_O_5_ 6%	4.0 μm **	No
Ba glass microfillers	Commercial/Inert	Schott, Mainz, Germany GM27884	SiO_2_ 55.0% BaO 25.0% B_2_O_3_ 10.0% Al_2_O_3_ 10.0%	1.0 μm **	Yes *** 3.20%
Silica nanofillers	Commercial/Inert	Evonik Degussa, Hanau, Germany Aerosil DT	SiO_2_ > 99.8%	12 nm	Yes *** 4–6%

* Composition determined by Inductively coupled plasma atomic emission spectroscopy (ICP-AES), analysis and recalculated to wt% (data from Zheng et al. [31]). ** d99 for 45S5 BG (G018-144) is 13 μm; d99 for Ba glass microfillers (GM27884) is 4 μm. *** Silanization performed by the manufacturers.

**Table 3 pharmaceutics-14-02241-t003:** Certified values for TMDW.

B (µg/L)	Ca (µg/L)	Cu (µg/L)
150 ± 3	31,000 ± 310	20.0 ± 0.2

## Data Availability

The datasets generated during and analyzed during the current study are available from the corresponding author upon reasonable request.

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
