# Peer review of "Using Copper-Doped Mesoporous Bioactive Glass Nanospheres to Impart Anti-Bacterial Properties to Dental Composites"

_pharmaceutics, 2022, doi:10.3390/pharmaceutics14102241_

Round 1

Reviewer 1 Report

The authors are requested to:

1- Describe in details the composite manufacturing steps and the methods of incorporating the fillers in the resin matrix, methods of silanization, if any preservatives were added,  effect of the added fillers on the shelf life of composite etc...  Full description of the manufacturer's details for the equipment used should be included.

2- Provide an illustrated figure for the experimental steps with special attention to the micro ct section describing the composite shrinkage as it is completely unclear in the manuscript.

3- Utilize  updated references discussing the potential of using the 45S5 bioglass as a remineralizing agent for enamel and dentin.

Reviewer 2 Report

I read this manual with great interest, because I deal with the subject of composite materials on a daily basis. I will give you some advice, however, that before sending this article to the publishing house, ask someone outside your team  to read your work, then you would avoid many mistakes, e.g. materials and methods parts  after discussion? There are errors in the graphs. Good work interesting topic but the whole text needs to be rewritten.

Introduction

 Line 66

In addition, a low-viscosity polymer can penetrate the pores and improve mechanical bonding of the filler particles after polymerization- this sentence needs to be corrected. Already during the mixing process, the low-viscosity resins penetrate into the space of the porous glass. The polymerization cures them and creates a mechanical bond. Because in the previous form it is a bit unclear.

Line 91

binary filler blend consisting of Ba-glass microfillers as a base and either Cu-MBGN, silica nanofillers, or 45S5  BG with a total filler amount of 65 wt%-

From what I understand, each material contains 65% filler (volumetric or mass- question?) And the first material contained

-          Ba-glass microfillers, Cu-MBGN, silica nanofillers

Second

-          45S5?

If yes, we will not compare two kinds of mateirals in one test. One is a microfilled composite where it contains larger glass particles (were they silanized?) And microsilica. And the other is a macrophilic composite with large particles. It is known that the first will have better mechanical properties thanks to different types of filler. Macrophilic composites invented in the 1960s had a fracture resistance of 90MPa, and the present ones have 130MPa, of course if the filler is silanized. If no, the fracture resistance is around 50-60 MPa. please explain it.

For the lines 91-96

  I think it needs to be written differently. After all, we all know that scientific work consists of a number of parts:

introduction, material and methods, results, discussion of the literature.

Materials and methods are lacking here. Where, as we all know, you need to describe what was tested, how the samples were prepared, what raw materials were used, what equipment (manufacturer), process conditions etc.

  If it was published earlier Ok, give literature references, or I would like a short summary of previous articles, because reading only this article I am like an interested reader a bit lost.

  As for what samples have been prepared, it would be good to make a block diagram and present it there, or a graphic abstract of the whole work. Because to be honest I have already lost what were the samples, how many?

Authors perform statistical analysis. What program what type of analysis was used 1 way arnova? This is also missing from the materials and methods ... paragraph. And then it is used in the results

Line 98-99

Therefore, we hypothesized that incorporating Cu-MBGN or the combination of Cu-MBGN and silica nanofillers into 9the resin matrix could reduce polymerization shrinkage, achieve anti-bacterial properties, reduce biofilm formation, and promote remineralization-

this is a thesis which should be at the end of the introduction. The rest of the description should be part of material and methods after adding my suggestions

lines 99-105

In this study, the micromorphology and uniform dispersion of filler particles in the resulting experimental composites were evaluated by scanning electron microscopy with energy dispersive X-Ray analysis (SEM/EDX). The effect of resin composites with Cu-MBGN on polymerization (degree of conversion, volumetric shrinkage, porosity) was investigated using attenuated total reflectance accessory on a Fourier transform infrared spectroscopy (ATR-FTIR) and micro computed tomography (micro-CT). Remineralizing and anti-bacterial properties were estimated from the release of calcium and copper ions by Inductively coupled plasma mass spectrometry (ICP-MS), while biofilm formation and anti-bacterial response to free-floating cariogenic bacteria were investigated using antibacterial assays.

Here, it would be necessary to add the manufacturers and the country of origin of the devices used for these investments. In my opinion, the brief description of the individual study was also good.

  Not everyone knows what such testing looks like. And when we publish, we would like our works to be cited as much as possible. If I read the article as a reader and I do not know what was done, I rather stop reading and I certainly will not quote this article in my work. please think about it?

Line 116

HEPES-buffered- what does this abbreviation mean, what was the composition of the buffer? Missing from the description of material and methods? Jelsi was used earlier in your research is a literature reference.

Figure 1

please enlarge the squares describing the individual elements. Even in zoom 150% you cannot see exactly what is Si and What is Ca. If it is not possible, then in the description below the figure, add the red color means phosphorus ions, green Cu but note figure 10 CuBG is copper green, but 10 Si copper is already blue and Zn is green. So no match. Please correct it.

Perhaps in this performance it would be worth going to work

Raszewski, Z.; Brzakalski, D.; Jałbrzykowski, M.; Pakuła, D.; Frydrych, M.; Przekop, R.E. Novel Multifunctional Spherosilicate-Based Coupling Agents for Improved Bond Strength and Quality in Restorative Dentistry. Materials 2022, 15, 3451. https://doi.org/10.3390/ ma15103451

Figure 2

15 Si again, there are different colors for zinc and Cu in between than in the photos presented above.

Line 156

The composites were hardened and their shrinkage was measured, Ok sure, but it is influenced by the type of festival source, the power of the device - lamps, exposure time, etc.

Was it similar to that in this article?

Sun J, Lin-Gibson S. X-ray microcomputed tomography for measuring polymerization shrinkage of polymeric dental composites. Dent Mater. 2008 Feb;24(2):228-34. doi: 10.1016/j.dental.2007.05.001. Epub 2007 Jun 18. PMID: 17572484.

Figure 5. Representative micro-CT images and reconstruction of experimental composites containing binary and ternary fillers. (A, D, G, J) shows composite specimens with overlaid dimensional changes before and after polymerization- what is before cuirng and  after curing (AD or GJ?, labeled micro-CT image overview. (B, E, H, K) represents the crossectional images, showing the pre- and post-polymerization areas of composites given in A, D, G, J. The crossectional images of composites (given in A, D, G, J) that show the dimensions of the samples before (red) and after (blue) polymerization are represented in C, F, I, L-

So the CFIL pictures were created with the superimposition of two pictures Before and after curing?

Figure 6

The graphs show the content of individual elements in calcium phosphorus in composites, etc., but the description and references from the text talks about porosity. I think there is a mistake here? Please correct this

Line 276

We investigated the anti-bacterial effect of the composites and biofilm formation by Streptococcus mutans (S. mutans) and Actinomyces naeslundii (A. naeslundii) - where the strains of these bacteria came from, how the tests were carried out, on a certain medium for how long, whether the contact method or extracts from the buffer was used this has to be described in   materials and methods sections

Discussion

. Well-blended and uniformly distributed fillers are a prerequisite for any experimental composite, especially if nanoscale fillers are included in the composition- A fair statement, however, you will not describe in this work how the material was mixed.

Line 334 -335

Agglomerates visible in 10-BG are the only non-uniform surface morphology. Agglomerates of 5-10 µm (Figure 1B) with high Ca and P content are easily recognized as 45S5 glass particles. Their d50 diameter is 4 µm but can reach 13 µm (d99).- If you claim that these are glass  45S5particles, then whether its granulation was measurable before the experiment or are these data from the literature, please explain. And maybe they were other agglomerates that were not mixed up during the preparation of the sample?

Materials and Methods- line 440

Sample preparation line 441.

Please describe whether the material samples were mixed by hand (then we cannot talk about their uniform mixing) or using a Producer mixer, mixing time, rotation?

literature

Xavier, T.A.; FRÓES-SALGADO, N.R.d.G.; Meier, M.M.; Braga, R.R. Influence of silane content and filler distribution on chemical-mechanical properties of resin composites. Brazilian Oral Research 2015, 29, 1-8- small letters

please move the material and methods to the appropriate place

good luck in further research

Round 2

Reviewer 2 Report

The work was tidied up and corrected, my concerns about it were clearly resolved. Thank you to the authors and I wish you further fruitful research